# CoCa: Contrastive Captioners are Image-Text Foundation Models

**Jiahui Yu**[⋆]                                                                *jiahuiyu@google.com*

**Zirui Wang**[⋆]                                                               *ziruiw@google.com*

**Vijay Vasudevan**

**Legg Yeung**

**Mojtaba Seyedhosseini**

**Yonghui Wu**

*Google Research*
[⋆] *Equal contribution.*

**Reviewed on OpenReview:** *https://openreview.net/forum?id=Ee277P3AYC*

## Abstract

Exploring large-scale pretrained foundation models is of significant interest in computer vision because these models can be quickly transferred to many downstream tasks. This paper presents **C**ontrastive **Ca**ptioner (**CoCa**), a minimalist design to pretrain an image-text encoder-decoder foundation model jointly with contrastive loss and captioning loss, thereby subsuming model capabilities from contrastive approaches like CLIP and generative methods like SimVLM. In contrast to standard encoder-decoder transformers where all decoder layers attend to encoder outputs, CoCa omits cross-attention in the first half of decoder layers to encode *unimodal* text representations, and cascades the remaining decoder layers which cross-attend to the image encoder for *multimodal* image-text representations. We apply a contrastive loss between unimodal image and text embeddings, in addition to a captioning loss on the multimodal decoder outputs which predicts text tokens autoregressively. By sharing the same computational graph, the two training objectives are computed efficiently with minimal overhead. CoCa is pretrained end-to-end and from scratch on both web-scale alt-text data and annotated images by treating all labels simply as text, seamlessly unifying natural language supervision for representation learning. Empirically, CoCa achieves state-of-the-art performance with zero-shot transfer or minimal task-specific adaptation on a broad range of downstream tasks, spanning visual recognition (ImageNet, Kinetics-400/600/700, Moments-in-Time), crossmodal retrieval (MSCOCO, Flickr30K, MSR-VTT), multimodal understanding (VQA, SNLI-VE, NLVR2), and image captioning (MSCOCO, NoCaps). Notably on ImageNet classification, CoCa obtains 86.3% *zero-shot* top-1 accuracy, 90.6% with a *frozen encoder* and learned classification head, and 91.0% with a *finetuned encoder*.

## 1 Introduction

Deep learning has recently witnessed the rise of foundation language models (Bommasani et al., 2021) such as BERT (Devlin et al., 2018), T5 (Raffel et al., 2019), GPT-3 (Brown et al., 2020), where models are pretrained on web-scale data and demonstrate generic multi-tasking capabilities through zero-shot, few-shot or transfer learning. Compared with specialized individual models, pretraining foundation models for massive downstream

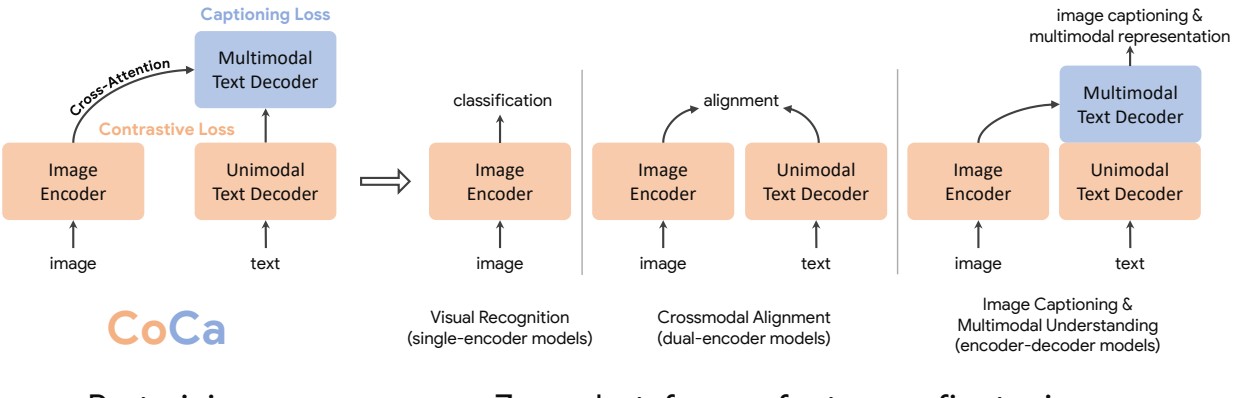

Figure 1: Overview of Contrastive Captioners (CoCa) pretraining as image-text foundation models. The pretrained CoCa can be used for downstream tasks including visual recognition, vision-language alignment, image captioning and multimodal understanding with zero-shot transfer, frozen-feature evaluation or end-to-end finetuning.

tasks can amortize training costs, providing opportunities to push the limits of model scale (Barham et al., 2022) for human-level intelligence.

For vision and vision-language problems, several foundation model candidates have been explored: (1) Pioneering works (Girshick et al., 2014; Long et al., 2015; Simonyan & Zisserman, 2014) have shown the effectiveness of **single-encoder models** pretrained with cross-entropy loss on image classification datasets such as ImageNet (Deng et al., 2009). The image encoder provides generic *visual representations* that can be adapted for various downstream tasks including image and video understanding (Dai et al., 2021; Zhang et al., 2021a). However, these models rely heavily on image annotations as labeled vectors and do not bake in knowledge of free-form human natural language, hindering their application to downstream tasks that involving both vision and language modalities. (2) Recently, a line of research (Radford et al., 2021; Jia et al., 2021; Yuan et al., 2021) has shown the feasibility of image-text foundation model candidates by pretraining two parallel encoders with a contrastive loss on web-scale noisy image-text pairs. In addition to the visual embeddings for vision-only tasks, the resulting **dual-encoder models** can additionally encode textual embeddings to the same latent space, enabling new *crossmodal alignment* capabilities such as zero-shot image classification and image-text retrieval. Nonetheless, these models are not directly applicable for joint vision-language understanding tasks such as visual question answering (VQA), due to missing joint components to learn fused image and text representations. (3) Another line of research (Vinyals et al., 2015; Wang et al., 2021b; 2022) has explored generative pretraining with **encoder-decoder models** to learn generic vision and multimodal representations. During pretraining, the model takes images on the encoder side and applies Language Modeling (LM) loss (or PrefixLM (Raffel et al., 2019; Wang et al., 2021b)) on the decoder outputs. For downstream tasks, the decoder outputs can then be used as joint representations for *multimodal understanding* tasks. While superior vision-language results (Wang et al., 2021b) have been attained with pretrained encoder-decoder models, they do not produce text-only representations aligned with image embeddings, thereby being less feasible and efficient for crossmodal alignment tasks.

In this work, we unify *single-encoder*, *dual-encoder* and *encoder-decoder* paradigms, and train one image-text foundation model that subsumes the capabilities of all three approaches. We propose a simple model family named Contrastive Captioners (CoCa) with a modified encoder-decoder architecture trained with both contrastive loss and captioning (generative) loss. As shown in Figure 1, we decouple the decoder transformer into two parts, a unimodal decoder and a multimodal decoder. We omit cross-attention in unimodal decoder layers to encode text-only representations, and cascade multimodal decoder layers cross-attending to image encoder outputs to learn *multimodal* image-text representations. We apply both the contrastive objective between outputs of the image encoder and unimodal text decoder, and the captioning objective at the

output of the multimodal decoder. Furthermore, CoCa is trained on both image annotation data and noisy image-text data by treating all labels simply as text. The generative loss on image annotation text provides a fine-grained training signal similar to the single-encoder cross-entropy loss approach, effectively subsuming all three pretraining paradigms into a single unified method.

The design of CoCa leverages contrastive learning for learning global representations and captioning for fine-grained region-level features, thereby benefiting tasks across all three categories shown in Figure 1. CoCa shows that a single pretrained model can outperform many specialized models using zero-shot transfer or minimal task-specific adaptation. For example, CoCa obtains 86.3% zero-shot accuracy on ImageNet and better zero-shot crossmodal retrieval on MSCOCO and Flickr30k. With a frozen-encoder, CoCa achieves 90.6% on ImageNet classification, 88.0%/88.5%/81.1% on Kinetics-40/600/700 and 47.4% on Moments-in-Time. After lightweight finetuning, CoCa further achieves 91.0% on ImageNet, 82.3% on VQA and 120.6 CIDEr score on NoCaps.

## 2 Related Work

**Vision Pretraining.** Pretraining ConvNets (Krizhevsky et al., 2012) or Transformers (Vaswani et al., 2017) on large-scale annotated data such as ImageNet (Girshick et al., 2014; Long et al., 2015; Simonyan & Zisserman, 2014), Instagram (Mahajan et al., 2018) or JFT (Zhai et al., 2021a) has become a popular strategy towards solving visual recognition problems including classification, localization, segmentation, video recognition, tracking and many other problems. Recently, self-supervised pretraining approaches have also been explored. BEiT (Bao et al., 2021) proposes a masked image modeling task following BERT (Devlin et al., 2018) in natural language processing, and uses quantized visual token ids as prediction targets. MAE (He et al., 2021) and SimMIM (Xie et al., 2021) remove the need for an image tokenizer and directly use a light-weight decoder or projection layer to regress pixel values. Nonetheless, these methods only learn models for the vision modality and thus they are not applicable to tasks that require joint reasoning over both image and text inputs.

**Vision-Language Pretraining.** In recent years, rapid progress has been made in vision-language pretraining (VLP), which aims to jointly encode vision and language in a fusion model. Early work (e.g. LXMERT (Tan & Bansal, 2019), UNITER (Chen et al., 2020), VinVL (Zhang et al., 2021b), VL-T5 (Cho et al., 2021)) in this direction relies on pretrained object detection modules such as Fast(er) R-CNN (Ren et al., 2015) to extract visual representations. Later efforts such as ViLT (Kim et al., 2021) and VLMo (Wang et al., 2021a) unify vision and language transformers, and train a multimodal transformer from scratch. More recently, a line of work has also explored zero-shot/few-shot learning for vision-language tasks by re-using pretrained large language models (Yang et al., 2022b; Jin et al., 2021; Tsimpoukelli et al., 2021). Compared to prior methods, this paper focuses on training a unified model from scratch subsuming the capability of multimodal understanding and generation.

**Image-Text Foundation Models.** Recent work has proposed image-text foundation models that can subsume both vision and vision-language pretraining. CLIP (Radford et al., 2021) and ALIGN (Jia et al., 2021) demonstrate that dual-encoder models pretrained with contrastive objectives on noisy image-text pairs can learn strong image and text representations for crossmodal alignment tasks and zero-shot image classification. Florence (Yuan et al., 2021) further develops this method with unified contrastive objective (Yang et al., 2022a), training foundation models that can be adapted for a wide range of vision and image-text benchmarks. To further improve zero-shot image classification accuracy, LiT (Zhai et al., 2021b) and BASIC (Pham et al., 2021a) first pretrain model on an large-scale image annotation dataset with cross-entropy and further finetune with contrastive loss on an noisy alt-text image dataset. Another line of research (Wang et al., 2021b; 2022; Piergiovanni et al., 2022) proposes encoder-decoder models trained with generative losses and shows strong results in vision-language benchmarks while the visual encoder still performs competitively on image classification. In this work, we focus on training an image-text foundation model from scratch in a single pretraining stage to unify these approaches. While recent works (Singh et al., 2021; Li et al., 2021; 2022) have also explored image-text unification, they require multiple pretraining stages of unimodal and multimodal modules to attain good performance. For example, ALBEF (Li et al., 2021) combines contrastive loss with masked language modelling (MLM) with a dual-encoder design. However, our approach is simpler

and more efficient to train while also enables more model capabilities: (1) CoCa only performs one forward and backward propagation for a batch of image-text pairs while ALBEF requires two (one on corrupted inputs and another without corruption), (2) CoCa is trained from scratch on the two objectives only while ALBEF is initialized from pretrained visual and textual encoders with additional training signals including momentum modules. (3) The decoder architecture with generative loss is preferred for natural language generation and thus directly enables image captioning.

## 3 Approach

We begin with a review of three foundation model families that utilize *natural language supervision* differently: single-encoder classification pretraining, dual-encoder contrastive learning, and encoder-decoder image captioning. We then introduce Contrastive Captioners (CoCa) that share the merits of both contrastive learning and image-to-caption generation under a simple architecture. We further discuss how CoCa models can quickly transfer to downstream tasks with zero-shot transfer or minimal task adaptation.

### 3.1 Natural Language Supervision

**Single-Encoder Classification.** The classic single-encoder approach pretrains a visual encoder through image classification on a large crowd-sourced image annotation dataset (*e.g.*, ImageNet (Deng et al., 2009), Instagram (Mahajan et al., 2018) or JFT (Zhai et al., 2021a)), where the vocabulary of annotation texts is usually fixed. These image annotations are usually mapped into discrete class vectors to learn with a cross-entropy loss as

$$\mathcal{L}_{\text{Cls}} = -p(y) \log q_\theta(x), \tag{1}$$

where $p(y)$ is a one-hot, multi-hot or smoothed label distribution from ground truth label $y$. The learned image encoder is then used as a generic visual representation extractor for downstream tasks.

**Dual-Encoder Contrastive Learning.** Compared to pretraining with single-encoder classification, which requires human-annotated labels and data cleaning, the dual-encoder approach exploits noisy web-scale text descriptions and introduces a learnable text tower to encode free-form texts. The two encoders are jointly optimized by contrasting the paired text against others in the sampled batch:

$$\mathcal{L}_{\text{Con}} = -\frac{1}{N} \Big( \underbrace{\sum_i^N \log \frac{\exp(x_i^\top y_i / \sigma)}{\sum_{j=1}^N \exp(x_i^\top y_j / \sigma)}}_{\text{image-to-text}} + \underbrace{\sum_i^N \log \frac{\exp(y_i^\top x_i / \sigma)}{\sum_{j=1}^N \exp(y_i^\top x_j / \sigma)}}_{\text{text-to-image}} \Big), \tag{2}$$

where $x_i$ and $y_j$ are normalized embeddings of the image in the $i$-th pair and that of the text in the $j$-th pair. $N$ is the batch size, and $\sigma$ is the temperature to scale the logits. In addition to the image encoder, the dual-encoder approach also learns an aligned text encoder that enables crossmodal alignment applications such as image-text retrieval and zero-shot image classification. Empirical evidence shows zero-shot classification is more robust (Radford et al., 2021; Jia et al., 2021; Andreassen et al., 2021) on corrupted or out-of-distribution images.

**Encoder-Decoder Captioning.** While the dual-encoder approach encodes the text as a whole, the generative approach (*a.k.a.* captioner) aims for detailed granularity and requires the model to predict the exact tokenized texts of $y$ autoregressively. Following a standard encoder-decoder architecture, the image encoder provides latent encoded features (*e.g.*, using a Vision Transformer (Dosovitskiy et al., 2021) or ConvNets (He et al., 2016)) and the text decoder learns to maximize the conditional likelihood of the paired text $y$ under the forward autoregressive factorization:

$$\mathcal{L}_{\text{Cap}} = -\sum_{t=1}^T \log P_\theta(y_t | y_{<t}, x). \tag{3}$$

The encoder-decoder is trained with teacher-forcing (Williams & Zipser, 1989) to parallelize computation and maximize learning efficiency. Unlike prior methods, the captioner approach yields a joint image-text representation that can be used for vision-language understanding, and is also capable of image captioning applications with natural language generation.

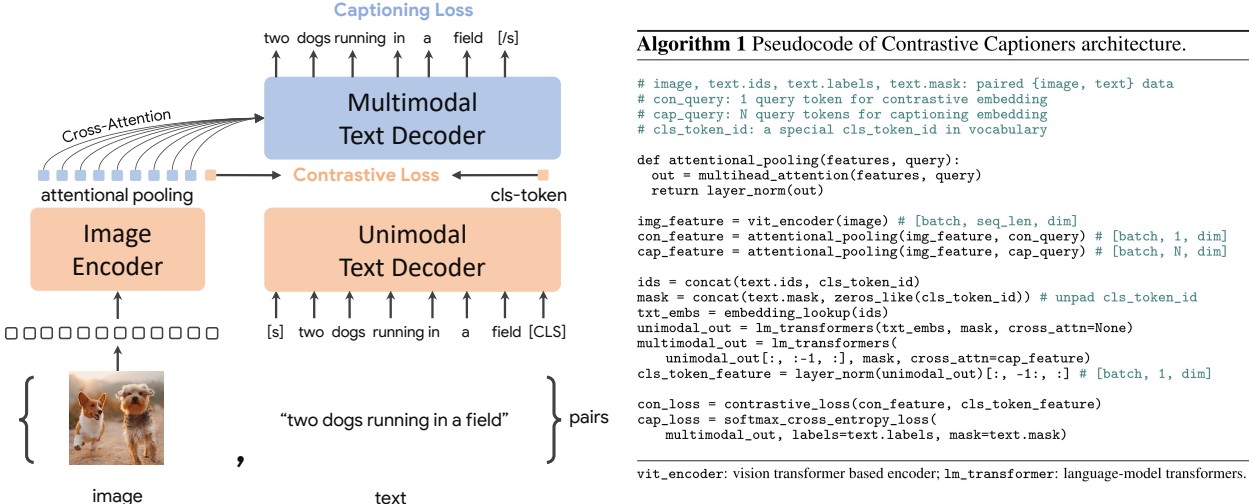

Figure 2: Detailed illustration of CoCa architecture and training objectives.

## 3.2 Contrastive Captioners Pretraining

Figure 2 depicts the proposed contrastive captioner (CoCa): a simple encoder-decoder approach that seamlessly combines the three training paradigms. Similar to standard image-text encoder-decoder models, CoCa encodes images to latent representations by a neural network encoder, for example, vision transformer (ViT) (Dosovitskiy et al., 2021) (used by default; it can also be other image encoders like ConvNets (He et al., 2016)), and decodes texts with a causal masking transformer decoder. Unlike standard decoder transformers, CoCa omits cross-attention in the first half of the decoder layers to encode *unimodal* text representations, and cascades the rest of the decoder layers, cross-attending to the image encoder for *multimodal* image-text representations. As a result, the CoCa decoder simultaneously produces both unimodal and multimodal text representations that allow us to apply both contrastive and generative objectives as

$$\mathcal{L}_{\text{CoCa}} = \lambda_{\text{Con}} \cdot \mathcal{L}_{\text{Con}} + \lambda_{\text{Cap}} \cdot \mathcal{L}_{\text{Cap}}, \tag{4}$$

where $\lambda_{\text{Con}}$ and $\lambda_{\text{Cap}}$ are loss weighting hyper-parameters. We note that the single-encoder cross-entropy classification objective can be interpreted as a special case of the generative approach applied on image annotation data, when the vocabulary is the set of all label names.

**Decoupled Text Decoder and CoCa Architecture.** The captioning approach optimizes the conditional likelihood of text while the contrastive approach uses an unconditional text representation. To address this dilemma and combine these two methods into a single model, we propose a simple *decoupled decoder* design where we split the decoder into unimodal and multimodal components, by skipping the cross-attention mechanism in the unimodal decoder layers. That is, the bottom $n_{\text{uni}}$ unimodal decoder layers encode the input text as latent vectors with causally-masked self-attention, and the top $n_{\text{multi}}$ multimodal layers further apply causally-masked self-attention and together with cross-attention to the output of the visual encoder. All decoder layers prohibit tokens from attending to future tokens, and it is straightforward to use the multimodal text decoder output for the captioning objective $\mathcal{L}_{\text{Cap}}$. For the contrastive objective $\mathcal{L}_{\text{Con}}$, we append a learnable [CLS] token at the end of the input sentence and use its corresponding output of unimodal decoder as the text embedding. We split the decoder in half such that $n_{\text{uni}} = n_{\text{multi}}$. Following ALIGN (Jia et al., 2021), we pretrain with image resolution of 288×288 and patch size 18×18, resulting in a total of 256 image tokens. Our largest CoCa model ("CoCa" in short) follows the ViT-giant setup in Zhai et al. (2021a) with 1B-parameters in the image encoder and 2.1B-parameters altogether with the text decoder. We also explore two smaller variants of "CoCa-Base" and "CoCa-Large" detailed in Table 1.

**Attentional Poolers.** It is noteworthy that the contrastive loss uses a single embedding for each image while the decoder usually attends to a sequence of image output tokens in an encoder-decoder captioner

(Wang et al., 2021b). Our preliminary experiments show that a single pooled image embedding helps visual recognition tasks as a global representation, while more visual tokens (thus more fine-grained) are beneficial for multimodal understanding tasks which require region-level features. Hence, CoCa adopts task-specific attentional pooling (Lee et al., 2019) to customize visual representations to be used for different types of training objectives and downstream tasks. Here, a *pooler* is a single multi-head attention layer with $n_{\text{query}}$ learnable queries, with the encoder output as both keys and values. Through this, the model can learn to pool embeddings with different lengths for the two training objectives, as shown in Figure 2. The use of task-specific pooling not only addresses different needs for different tasks but also introduces the pooler as a natural task adapter. We use attentional poolers in pretraining for generative loss $n_{\text{query}} = 256$ and contrastive loss $n_{\text{query}} = 1$.

**Pretraining Efficiency.** A key benefit of the decoupled autoregressive decoder design is that it can compute two training losses considered efficiently. Since unidirectional language models are trained with causal masking on complete sentences, the decoder can efficiently generate outputs for both contrastive and generative losses with a single forward propagation (compared to two passes for a bidirectional approach (Li et al., 2021)). Therefore, the majority of the compute is shared between the two losses and CoCa only induces minimal overhead compared to standard encoder-decoder models. On the other hand, while many existing methods (Zhai et al., 2021b; Pham et al., 2021a; Singh et al., 2021; Wang et al., 2021a; Li et al., 2021; 2022) train model components with multiple stages on various data sources and/or modalities, CoCa is pretrained end-to-end from scratch directly with various data sources (*i.e.*, annotated images and noisy alt-text images) by treating all labels as texts for both contrastive and generative objectives.

### 3.3 Contrastive Captioners for Downstream Tasks

**Zero-shot Transfer.** A pretrained CoCa model performs many tasks in a zero-shot manner by leveraging both image and text inputs, including zero-shot image classification, zero-shot image-text cross-retrieval, zero-shot video-text cross-retrieval. Following previous practices (Radford et al., 2021; Zhai et al., 2021b), "zero-shot" here is different from classical zero-shot learning in that during pretraining, the model may see relevant supervised information, but no supervised examples are used during the transfer protocol. For the pretraining data, we follow strict de-duplication procedures introduced in Jia et al. (2021); Zhai et al. (2021b) to filter all near-domain examples to our downstream tasks.

**Frozen-feature Evaluation.** As discussed in the previous section, CoCa adopts task-specific attentional pooling (Lee et al., 2019) (*pooler* for brevity) to customize visual representations for different types downstream tasks while sharing the backbone encoder. This enables the model to obtain strong performance as a *frozen encoder* where we only learn a new pooler to aggregate features. It can also benefit to multi-task problems that share the same frozen image encoder computation but different task-specific heads. As also discussed in He et al. (2021), linear-evaluation struggles to accurately measure learned representations and we find the attentional poolers are more practical for real-world applications.

**CoCa for Video Action Recognition.** We use a simple approach to enable a learned CoCa model for video action recognition tasks. We first take multiple frames of a video and feed each frame into the *shared* image encoder individually as shown in Figure 3. For frozen-feature evaluation or finetuning, we learn an additional pooler on top of the spatial and temporal feature tokens with a softmax cross-entropy loss. Note the pooler has a single query token thus the computation of pooling over all spatial and temporal tokens is not expensive. For zero-shot video-text retrieval, we use an even simpler approach by computing the mean embedding of 16 frames of the video (frames are uniformly sampled from a video). We also encode the captions of each video as target embeddings when computing retrieval metrics (similar to the image-text case).

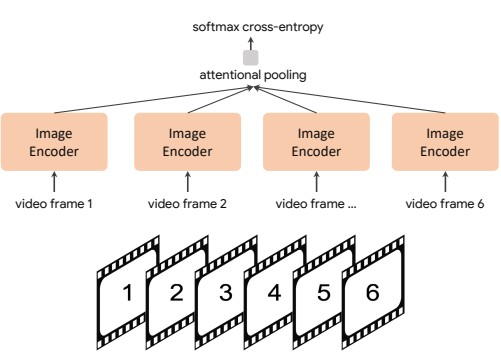

Figure 3: CoCa for video recognition.

| Model | Image Encoder | | | Text Decoder | | | | Image / Text | | |
|---|---|---|---|---|---|---|---|---|---|---|
| | Layers | MLP | Params | $n_\text{uni}$ | $n_\text{multi}$ | MLP | Params | Hidden | Heads | Total Params |
| CoCa-Base | 12 | 3072 | 86M | 12 | 12 | 3072 | 297M | 768 | 12 | 383M |
| CoCa-Large | 24 | 4096 | 303M | 12 | 12 | 4096 | 484M | 1024 | 16 | 787M |
| **CoCa** | 40 | 6144 | 1B | 18 | 18 | 5632 | 1.1B | 1408 | 16 | 2.1B |

Table 1: Size variants of CoCa. Both image encoder and text decoder are Transformers (Dosovitskiy et al., 2021; Vaswani et al., 2017).

## 4 Experiments

In this section, we first describe the details of our experimental setup. The main results are presented next organized as visual recognition tasks, crossmodal alignment tasks, image captioning and multimodal understanding tasks. Our main results are conducted under three categories for downstream tasks: zero-shot transfer, frozen-feature evaluation and finetuning. We also present ablation experiments including training objectives and architecture designs.

### 4.1 Training Setup

**Data.** As discussed in Section 3.2, CoCa is pretrained from scratch in a single stage on both web-scale alt-text data and annotated images by treating all labels simply as texts. We use the JFT-3B dataset (Zhai et al., 2021a) with label names as the paired texts, and the ALIGN dataset (Jia et al., 2021) with noisy alt-texts. Similar to Pham et al. (2021a), we randomly shuffle and concatenate label names of each image in JFT together with a prompt sampled from Radford et al. (2021). An example of the resulting text label of a JFT image would look like "a photo of the cat, animal". Unlike prior models (Zhai et al., 2021b; Pham et al., 2021a) that also use the combination of these two datasets, we train all model parameters from scratch at the same time without pretraining an image encoder with supervised cross-entropy loss for simplicity and pretraining efficiency. To ensure fair evaluation, we follow the strict de-duplication procedures introduced in (Zhai et al., 2021b; Jia et al., 2021) to filter all near-domain examples (3.6M images are removed in total) to our downstream tasks. To tokenize text input, we use a sentence-piece model (Sennrich et al., 2015; Kudo, 2018) with a vocabulary size of 64k trained on the sampled pretraining dataset.

**Optimization.** Our models are implemented in the Lingvo framework (Shen et al., 2019) with GSPMD (Huang et al., 2019; Xu et al., 2020; Lepikhin et al., 2020; Xu et al., 2021) for scaling performance. Following (Pham et al., 2021a), we use a batch size of 65,536 image-text pairs, where half of each batch comes from JFT and ALIGN, respectively. All models are trained on the combined contrastive and captioning objectives in Eq.(4) for 500k steps, roughly corresponding to 5 epochs on JFT and 10 epochs on ALIGN. As shown later in our studies, we find a larger captioning loss weight is better and thus $\lambda_\text{Cap} = 2.0$ and $\lambda_\text{Con} = 1.0$. Following Jia et al. (2021), we apply a contrastive loss with a trainable temperature $\tau$ with an initial value of 0.07. For memory efficiency, we use the Adafactor (Shazeer & Stern, 2018) optimizer with $\beta_1 = 0.9, \beta_2 = 0.999$ and decoupled weight decay (Loshchilov & Hutter, 2017) ratio of 0.01. We warm up the learning rate for the first 2% of training steps to a peak value of $8 \times 10^{-4}$, and linearly decay it afterwards. Pretraining CoCa takes about 5 days on 2,048 CloudTPUv4 chips. Following Radford et al. (2021); Jia et al. (2021); Yuan et al. (2021), we continue pretraining for one epoch on a higher resolution of $576 \times 576$. For finetuning evaluation, we mainly follow simple protocols and directly train CoCa on downstream tasks without further metric-specific tuning like CIDEr scores (details in Appendix A and B).

### 4.2 Main Results

We extensively evaluate the capabilities of CoCa models on a wide range of downstream tasks as a pretrained foundation model. We mainly consider core tasks of three categories that examine (1) visual recognition, (2) crossmodal alignment, and (3) image captioning and multimodal understanding capabilities. Since CoCa produces both aligned unimodal representations and fused multimodal embeddings at the same time, it is easily transferable to all three task groups with minimal adaption. Figure 4 summarizes the performance on key benchmarks of CoCa compared to other dual-encoder and encoder-decoder foundation models and

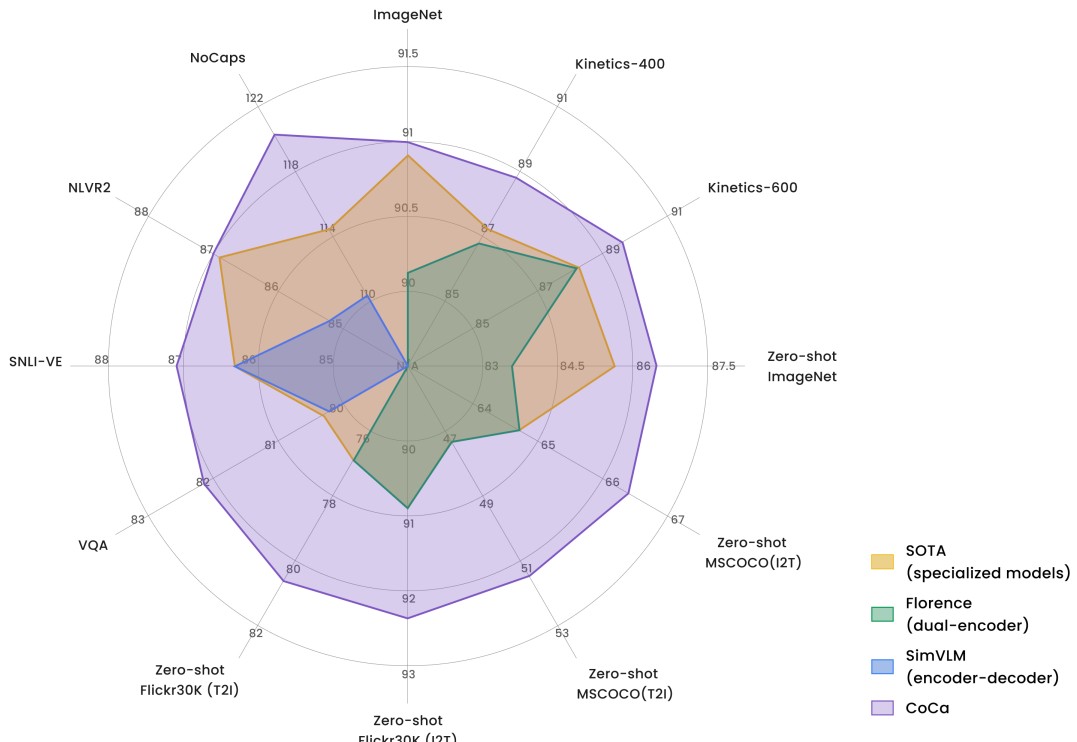

Figure 4: Comparison of CoCa with other image-text foundation models (without task-specific customization) and multiple state-of-the-art task-specialized models.

| Model | ImageNet |
|---|---|
| ALIGN[a] | 88.6 |
| Florence[b] | 90.1 |
| MetaPseudoLabels[c] | 90.2 |
| CoAtNet[d] | 90.9 |
| ViT-G[e] | 90.5 |
|   + Model Soups[f] | 90.9 |
| CoCa (frozen) | 90.6 |
| CoCa (finetuned) | **91.0** |

| Model | K-400 | K-600 | K-700 | Moments-in-Time |
|---|---|---|---|---|
| ViViT[g] | 84.8 | 84.3 | - | 38.0 |
| MoViNet[h] | 81.5 | 84.8 | 79.4 | 40.2 |
| VATT[i] | 82.1 | 83.6 | - | 41.1 |
| Florence[b] | 86.8 | 88.0 | - | - |
| MaskFeat[k] | 87.0 | 88.3 | 80.4 | |
| CoVeR[l] | 87.2 | 87.9 | 78.5 | 46.1 |
| CoCa (frozen) | 88.0 | 88.5 | 81.1 | 47.4 |
| CoCa (finetuned) | **88.9** | **89.4** | **82.7** | **49.0** |

Table 2: Image classification and video action recognition with frozen encoder or finetuned encoder. Model reference: [a](Jia et al., 2021) [b](Yuan et al., 2021) [c](Pham et al., 2021b) [d](Dai et al., 2021) [e](Zhai et al., 2021a) [g](Wortsman et al., 2022) [g](Arnab et al., 2021) [h](Kondratyuk et al., 2021) [i](Akbari et al., 2021) [k](Wei et al., 2021) [l](Zhang et al., 2021a).

state-of-the-art task-specialized methods. CoCa sets new state-of-the-art results on tasks of all three categories with a single pretrained checkpoint.

### 4.2.1   Visual Recognition Tasks

Our visual recognition experiments are conducted on ImageNet (Deng et al., 2009) as image recognition benchmark, and multiple video datasets including Kinetics-400 (Kay et al., 2017), Kinetics-600 (Carreira et al., 2018), Kinetics-700 (Carreira et al., 2019), Moments-in-Time (Monfort et al., 2019) as test-beds for video action recognition; it is noteworthy that CoCa pretrains on image data only, without accessing any extra

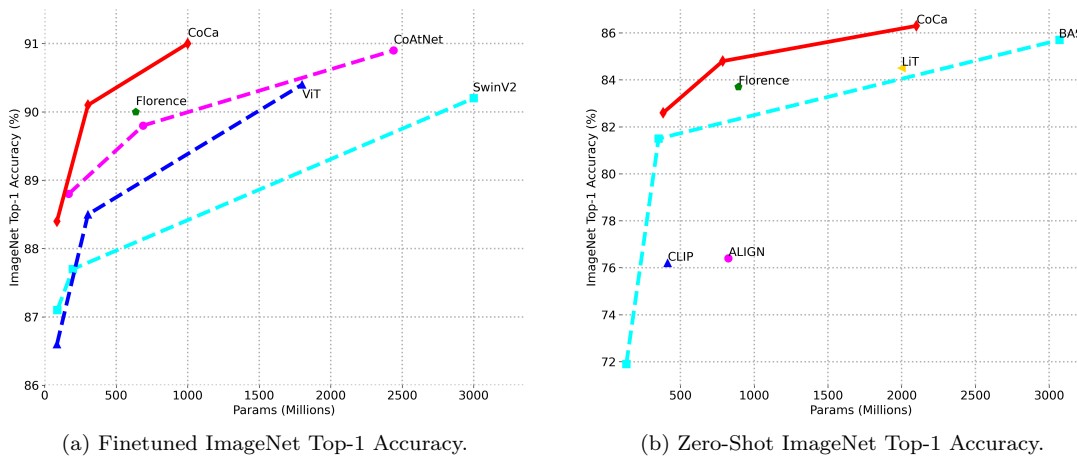

(a) Finetuned ImageNet Top-1 Accuracy.  (b) Zero-Shot ImageNet Top-1 Accuracy.

Figure 5: Image classification scaling performance of model sizes.

| | Flickr30K (1K test set) | | | | | | MSCOCO (5K test set) | | | | | |
| | Image → Text | | | Text → Image | | | Image → Text | | | Text → Image | | |
| Model | R@1 | R@5 | R@10 | R@1 | R@5 | R@10 | R@1 | R@5 | R@10 | R@1 | R@5 | R@10 |
|---|---|---|---|---|---|---|---|---|---|---|---|---|
| CLIP (Radford et al., 2021) | 88.0 | 98.7 | 99.4 | 68.7 | 90.6 | 95.2 | 58.4 | 81.5 | 88.1 | 37.8 | 62.4 | 72.2 |
| ALIGN (Jia et al., 2021) | 88.6 | 98.7 | 99.7 | 75.7 | 93.8 | 96.8 | 58.6 | 83.0 | 89.7 | 45.6 | 69.8 | 78.6 |
| FLAVA (Singh et al., 2021) | 67.7 | 94.0 | - | 65.2 | 89.4 | - | 42.7 | 76.8 | - | 38.4 | 67.5 | - |
| FILIP (Yao et al., 2021) | 89.8 | 99.2 | 99.8 | 75.0 | 93.4 | 96.3 | 61.3 | 84.3 | 90.4 | 45.9 | 70.6 | 79.3 |
| Florence (Yuan et al., 2021) | 90.9 | 99.1 | - | 76.7 | 93.6 | - | 64.7 | 85.9 | - | 47.2 | 71.4 | - |
| CoCa-Base | 89.8 | 98.8 | 99.8 | 76.8 | 93.7 | 96.8 | 63.8 | 84.7 | 90.7 | 47.5 | 72.4 | 80.9 |
| CoCa-Large | 91.4 | 99.2 | 99.9 | 79.0 | 95.1 | 97.4 | 65.4 | 85.6 | 91.4 | 50.1 | 73.8 | 81.8 |
| CoCa | **92.5** | **99.5** | **99.9** | **80.4** | **95.7** | **97.7** | **66.3** | **86.2** | **91.8** | **51.2** | **74.2** | **82.0** |

Table 3: Zero-shot image-text retrieval results on Flickr30K (Plummer et al., 2015) and MSCOCO (Chen et al., 2015) datasets.

video datasets. We apply the CoCa encoder on video frames individually (Section 3.3) without early fusion of temporal information, yet the resulting CoCa-for-Video model performs better than many spatio-temporal early-fused video models.

**Frozen-feature.** We apply a pretrained frozen CoCa model on both image classification and video action recognition. The encoder is used for both tasks while the decoder is discarded. As discussed in Section 3.3, an attentional pooling is learned together with a softmax cross-entropy loss layer on top of the embedding outputs from CoCa encoder. For video classification, a single query-token is learned to weight outputs of all tokens of spatial patches × temporal frames. We set a learning rate of $5 \times 10^{-4}$ on both attentional pooler and softmax, batch size of 128, and a cosine learning rate schedule (details in Appendix A). For video action recognition, we compare CoCa with other approaches on the same setup (*i.e.*, without extra supervised video data and without audio signals as model inputs). As shown in Table 2, without finetuning full encoder, CoCa already achieves competitive Top-1 classification accuracies compared to specialized image and outperforms prior best-performing specialized methods on video tasks.

**Finetuning.** Based on the architecture of frozen-feature evaluation, we further finetune CoCa encoders on image and video datasets individually with a smaller learning rate of $1 \times 10^{-4}$. More experimental details are summarized in the Appendix A. The finetuned CoCa has improved performance across these tasks. Notably, CoCa obtains 91.0% Top-1 accuracy on ImageNet, as well as better video action recognition results compared with recent video approaches. More importantly, CoCa models use much less parameters than other methods in the visual encoder as shown in Figure 5a. These results suggest the proposed framework efficiently combines text training signals and thus is able to learn high-quality visual representation better than the classical single-encoder approach.

| Model | ImageNet | ImageNet-A | ImageNet-R | ImageNet-V2 | ImageNet-Sketch | ObjectNet | Average |
|---|---|---|---|---|---|---|---|
| CLIP (Radford et al., 2021) | 76.2 | 77.2 | 88.9 | 70.1 | 60.2 | 72.3 | 74.3 |
| ALIGN (Jia et al., 2021) | 76.4 | 75.8 | 92.2 | 70.1 | 64.8 | 72.2 | 74.5 |
| FILIP (Yao et al., 2021) | 78.3 | - | - | - | - | - | - |
| Florence (Yuan et al., 2021) | 83.7 | - | - | - | - | - | - |
| LiT (Zhai et al., 2021b) | 84.5 | 79.4 | 93.9 | 78.7 | - | 81.1 | - |
| BASIC (Pham et al., 2021a) | 85.7 | 85.6 | 95.7 | 80.6 | 76.1 | 78.9 | 83.7 |
| CoCa-Base | 82.6 | 76.4 | 93.2 | 76.5 | 71.7 | 71.6 | 78.7 |
| CoCa-Large | 84.8 | 85.7 | 95.6 | 79.6 | 75.7 | 78.6 | 83.3 |
| CoCa | **86.3** | **90.2** | **96.5** | **80.7** | **77.6** | **82.7** | **85.7** |

Table 4: Zero-shot image classification results on ImageNet (Deng et al., 2009), ImageNet-A (Hendrycks et al., 2021b), ImageNet-R (Hendrycks et al., 2021a), ImageNet-V2 (Recht et al., 2019), ImageNet-Sketch (Wang et al., 2019) and ObjectNet (Barbu et al., 2019).

### 4.2.2 Crossmodal Alignment Tasks

Unlike other fusion-based foundation methods (Wang et al., 2021b; Singh et al., 2021; Wang et al., 2022), CoCa is naturally applicable to crossmodal alignment tasks since it generates aligned image and text unimodal embeddings (see Appendix C for results on video-text retrieval). In particular, we are interested in the zero-shot setting where all parameters are frozen after pretraining and directly used to extract embeddings. Here, we use the same embeddings used for contrastive loss during pretraining, and thus the multimodal text decoder is not used.

**Zero-Shot Image-Text Retrieval.** We evaluate CoCa on the two standard image-text retrieval benchmarks: MSCOCO (Chen et al., 2015) and Flickr30K (Plummer et al., 2015). Following the CLIP setting (Radford et al., 2021), we first independently feed each image/text to the corresponding encoder and obtain embeddings for all image/text in the test set. We then retrieve based on cosine similarity scores over the whole test set. As shown in Table 3, CoCa significantly improves over prior methods on both image-to-text and text-to-image retrievals on all metrics. In addition, our model is parameter-efficient, with CoCa-Base already outperforming strong baselines (CLIP (Radford et al., 2021) and ALIGN (Jia et al., 2021)) and CoCa-Large outperforming Florence (Yuan et al., 2021) (which contains a parameter count comparable to ViT-Huge). This shows that CoCa learns good unimodal representations *and* aligns them well across modalities.

**Zero-Shot Image Classification.** Following prior work (Radford et al., 2021; Jia et al., 2021), we use the aligned image/text embeddings to perform zero-shot image classification by matching images with label names without finetuning. We follow the exact setup in Radford et al. (2021) and apply the same set of prompts used for label class names. As shown in Table 4, CoCa sets new state-of-the-art zero-shot classification results on ImageNet. Notably, CoCa uses fewer parameters than prior best model (Pham et al., 2021a) while smaller CoCa variants already outperform strong baselines (Radford et al., 2021; Yuan et al., 2021), as shown in Figure 5b. In addition, our model demonstrates effective generalization under zero-shot evaluation, consistent with prior findings (Radford et al., 2021; Jia et al., 2021), with CoCa improving on all six datasets considered. Lastly, while prior models (Zhai et al., 2021b; Pham et al., 2021a) found sequentially pretraining with single-encoder and dual-encoder methods in multiple stages is crucial to performance gains, our results show it is possible to attain strong performance by unifying training objectives and datasets in a single-stage framework.

### 4.2.3 Image Captioning and Multimodal Understanding Tasks

Another key advantage of CoCa is its ability to process multimodal embeddings as an encoder-decoder model trained with the generative objective. Therefore, CoCa can perform both image captioning and multimodal understanding downstream tasks without any further fusion adaptation (Shen et al., 2021; Dou et al., 2021). Overall, experimental results suggest CoCa reaps the benefit of a encoder-decoder model to obtain strong multimodal understanding and generation capabilities, in addition to the vision and retrieval capabilities as a dual-encoder method.

| Model | VQA | | SNLI-VE | | NLVR2 | |
|---|---|---|---|---|---|---|
| | test-dev | test-std | dev | test | dev | test-p |
| UNITER (Chen et al., 2020) | 73.8 | 74.0 | 79.4 | 79.4 | 79.1 | 80.0 |
| VinVL (Zhang et al., 2021b) | 76.6 | 76.6 | - | - | 82.7 | 84.0 |
| CLIP-ViL (Shen et al., 2021) | 76.5 | 76.7 | 80.6 | 80.2 | - | - |
| ALBEF (Li et al., 2021) | 75.8 | 76.0 | 80.8 | 80.9 | 82.6 | 83.1 |
| BLIP (Li et al., 2022) | 78.3 | 78.3 | - | - | 82.2 | 82.2 |
| OFA (Wang et al., 2022) | 79.9 | 80.0 | 90.3[†] | 90.2[†] | - | - |
| VLMo (Wang et al., 2021a) | 79.9 | 80.0 | - | - | 85.6 | 86.9 |
| SimVLM (Wang et al., 2021b) | 80.0 | 80.3 | 86.2 | 86.3 | 84.5 | 85.2 |
| Florence (Yuan et al., 2021) | 80.2 | 80.4 | - | - | - | - |
| METER (Dou et al., 2021) | 80.3 | 80.5 | - | - | - | - |
| CoCa | **82.3** | **82.3** | **87.0** | **87.1** | **86.1** | **87.0** |

Table 5: Multimodal understanding results comparing vision-language pretraining methods. [†]OFA uses both image and text premises as inputs while other models utilize the image only.

| | MSCOCO | | | | NoCaps | | | |
|---|---|---|---|---|---|---|---|---|
| | | | | | Valid | | Test | |
| | B@4 | M | C | S | C | S | C | S |
| CLIP-ViL (Shen et al., 2021) | 40.2 | 29.7 | 134.2 | 23.8 | - | - | - | - |
| BLIP (Li et al., 2022) | 40.4 | - | 136.7 | - | 113.2 | 14.8 | - | - |
| VinVL(Zhang et al., 2021b) | 41.0 | 31.1 | 140.9 | 25.4 | 105.1 | 14.4 | 103.7 | 14.4 |
| SimVLM (Wang et al., 2021b) | 40.6 | 33.7 | 143.3 | **25.4** | 112.2 | - | 110.3 | 14.5 |
| LEMON (Hu et al., 2021) | **41.5** | 30.8 | 139.1 | 24.1 | 117.3 | 15.0 | 114.3 | 14.9 |
| LEMON$_{\text{SCST}}$ (Hu et al., 2021)[†] | 42.6 | 31.4 | 145.5 | 25.5 | - | - | - | - |
| OFA (Wang et al., 2022)[†] | 43.5 | 31.9 | 149.6 | 26.1 | - | - | - | - |
| CoCa | 40.9 | **33.9** | **143.6** | 24.7 | **122.4** | **15.5** | **120.6** | **15.5** |

Table 6: Image captioning results on MSCOCO and NoCaps (B@4: BLEU@4, M: METEOR, C: CIDEr, S: SPICE). [†]Models finetuned with CIDEr optimization.

**Multimodal Understanding.** As shown in Wang et al. (2021b), the output of encoder-decoder models can jointly encode image and text inputs, and can be used for tasks that require reasoning over both modalities. We consider three popular multimodal understaning benchmarks: visual question answering (VQA v2 (Goyal et al., 2017)), visual entailment (SNLI-VE (Xie et al., 2019)), and visual reasoning (NLVR2 (Suhr et al., 2018)). We mainly follow the settings in Wang et al. (2021b) and train linear classifiers on top of the decoder outputs to predict answers (more details in Appendix B). Our results in Table 5 suggest that CoCa outperforms strong vision-language pretraining (VLP) baselines and obtains the best performance on all three tasks. While prior dual-encoder models (Radford et al., 2021; Yuan et al., 2021) do not contain fusion layers and thus require an additional VL pretraining stage for downstream multimodal understanding tasks, CoCa subsumes the three pretraining paradigms and obtains better performance on VL tasks with lightweight finetuning.

**Image Captioning.** In addition to multimodal classification tasks, CoCa is also directly applicable to image captioning tasks as an encoder-decoder model. We finetune CoCa with the captioning loss $\mathcal{L}_{\text{Cap}}$ only on MSCOCO (Chen et al., 2015) captioning task and evaluate on both MSCOCO Karpathy-test split and NoCaps (Agrawal et al., 2019) online evaluation. As shown by experiments in Table 6, CoCa outperforms strong baselines trained with cross-entropy loss on MSCOCO, and achieves results comparable to methods with CIDEr metric-specific optimization (Rennie et al., 2017). It is noteworthy that we do not use CIDEr-specific optimization (Rennie et al., 2017) for simplicity. On the challenging NoCaps benchmark, CoCa obtains better results on both validation and test splits (generated examples shown in Figure 6). These results showcase the generative capability of CoCa as an image-text foundation model.

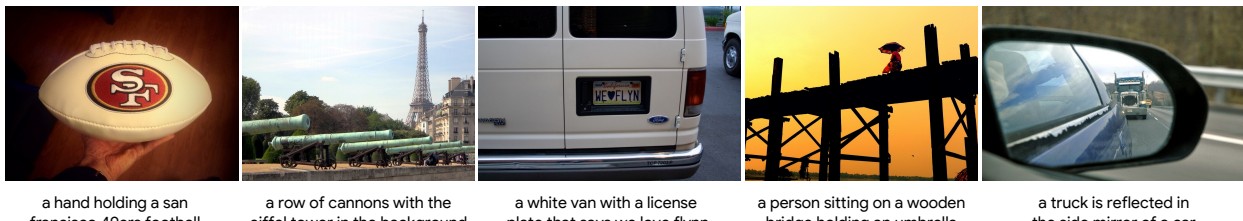

a hand holding a san francisco 49ers football

a row of cannons with the eiffel tower in the background

a white van with a license plate that says we love flynn

a person sitting on a wooden bridge holding an umbrella

a truck is reflected in the side mirror of a car

Figure 6: Curated samples of text captions generated by CoCa with NoCaps images as input.

| loss | LE | FT |
|---|---|---|
| $\mathcal{L}_{\text{Cls}}$ | 81.0 | 85.1 |
| $\mathcal{L}_{\text{Cap}}$ | 82.1 | 84.9 |

(a) Encoder-decoder *vs.* single-encoder models (trained on JFT).

| loss | ZS | VQA | TPU cost |
|---|---|---|---|
| $\mathcal{L}_{\text{Con}}$ | 70.7 | 59.2 | 1× |
| $\mathcal{L}_{\text{Cap}}$ | - | 68.9 | 1.17× |
| $\mathcal{L}_{\textbf{CoCa}}$ | **71.6** | **69.0** | **1.18×** |

(b) Training objectives ablation.

| $\lambda_{\text{Cap}} : \lambda_{\text{Con}}$ | ZS | VQA |
|---|---|---|
| 1:1 | 71.5 | 68.6 |
| 1:2 | 71.0 | 68.1 |
| **2:1** | **71.6** | **69.0** |

(c) Training objectives weights.

| $n_{\text{uni}}$ | ZS | VQA |
|---|---|---|
| 3 | 70.2 | 69.0 |
| **6** | **71.6** | **69.0** |
| 9 | 71.4 | 68.8 |

(d) Unimodal decoder layers.

| variant | AE | MSCOCO |
|---|---|---|
| **1 [CLS]** | **80.7** | **41.4** |
| + text tokens | 80.3 | 40.2 |
| 8 [CLS] | 80.3 | 36.9 |
| + text tokens | 80.4 | 40.3 |

(e) Contrastive text embedding design ablation.

| variant | ZS | VQA |
|---|---|---|
| parallel | 71.2 | 68.7 |
| **cascade** | **71.6** | **69.0** |
| $n_{\text{query}} = 0$ | 71.5 | 69.0 |
| $n_{\text{query}} = 1$ | 69.3 | 64.4 |
| $n_{\text{query}} = 32$ | 71.2 | 68.2 |

(f) Attentional pooler design ablation.

Table 7: CoCa ablation experiments. On ImageNet classification, we report top-1 accuracy for: zero-shot (ZS), linear evaluation (LE), attentional evaluation (AE) using pooler on frozen feature, and finetuning (FT). On MSCOCO retrieval, we report the average of image-to-text and text-to-image R@1. On VQA, we report the dev-set vqa score. The default CoCa setting is **bold**.

## 4.3 Ablation Analysis

We extensively ablate the properties of CoCa on a smaller model variant. Specifically, we train CoCa-Base with a reduced 12 decoder layers and a total batch size of 4,096. We mainly evaluate using zero-shot image classification and VQA, since the former covers both visual representation quality and crossmodal alignment, while the later is representative for multimodal reasoning.

**Captioning *vs.* Classification.** We first examine the effectiveness of captioning loss on image annotation datasets. To do this, we train a naive encoder-decoder model using $\mathcal{L}_{\text{Cap}}$ on the JFT-3B dataset, and compare with a standard ViT-Base single-encoder model trained with $\mathcal{L}_{\text{Cls}}$ in Table 7a. We find encoder-decoder models to perform on par with single-encoder pretraining on both linear evaluation and finetuned results. This suggests that the generative pretraining subsumes classification pretraining, consistent with our intuition that $\mathcal{L}_{\text{Cls}}$ is a special case of $\mathcal{L}_{\text{Cap}}$ when text vocabulary is the set of all possible class names. Thus, our CoCa model can be interpreted as an effective unification of the three paradigms. This explains why CoCa does not need a pretrained visual encoder to perform well.

**Training Objectives.** We study the effects of the two training objectives and compare CoCa with single-objective variants in Table 7b. Compared to the contrastive-only model, CoCa significantly improves both zero-shot alignment and VQA (notice that the contrastive-only model requires additional fusion for VQA). CoCa performs on par with the captioning-only model on VQA while it additionally enables retrieval-style tasks such as zero-shot classification. Table 7c further studies loss ratios and suggests that the captioning loss not only improves VQA but also zero-shot alignment between modalities. We hypothesize that generative objectives learn fine-grained text representations that further improve text understanding. Finally, we compare

training costs in Table 7b (measured in TPUv3-core-days; larger is slower) and find CoCa to be as efficient as the captioning-only model (*a.k.a.*naive encoder-decoder with same architecture as CoCa) due to the sharing of compute between two objectives. These suggest combining the two losses induces new capabilities and better performance with minimal extra cost.

**Unimodal Textual Representation.** CoCa introduces a novel decoder design and we ablate its components. In Table 7d, we vary the number of unimodal decoder layers (while keeping the total number of layers the same). Intuitively, fewer unimodal text layers leads to worse zero-shot classification due to lack of capacity for good unimodal text understanding, while fewer multimodal layers reduces the model's power to reason over multimodal inputs such as VQA. Overall, we find decoupling the decoder in half maintains a good balance. One possibility is that global text representation for retrieval doesn't require deep modules (Pham et al., 2021a) while early fusion for shallow layers may also be unnecessary for multimodal understanding. Another key design of unimodal textual representation is the application of [CLS] tokens. In particular, we experiment with the number of learnable [CLS] tokens as well as the aggregation design. For the later, we aggregate over either the [CLS] tokens only (denoted as N [CLS]) or the concatenation of [CLS] and the original input sentence (denoted as N [CLS] + text tokens). Interestingly, in Table 7e we find training a single [CLS] token without the original input is preferred for both vision-only and crossmodal retrieval tasks. This indicates that learning an additional simple sentence representation mitigates interference between contrastive and captioning loss, and is powerful enough for strong generalization.

**Attentional Poolers.** CoCa exploits attentional poolers in its design both for different pretraining objectives and objective-specific downstream task adaptations. In pretraining, we compare a few design variants on using poolers for contrastive loss and generative loss: (1) the "parallel" design which extracts both contrastive and generative losses at the same time on Vision Transformer encoder outputs as shown in Figure 2, and (2) the "cascade" design which applies the contrastive pooler on top of the outputs of the generative pooler. Table 7f shows the results of these variants. Empirically, we find at small scale the "cascade" version (contrastive pooler on top of the generative pooler) performs better and is used by default in all CoCa models. We also study the effect of number of queries where $n_{\text{query}} = 0$ means no generative pooler is used (thus all ViT output tokens are used for decoder cross-attention). Results show that both tasks prefer longer sequences of detailed image tokens at a cost of slightly more computation and parameters. As a result, we use a generative pooler of length 256 to improve multimodal understanding benchmarks while still maintaining the strong frozen-feature capability.

## 5 Conclusion

In this work we present Contrastive Captioners (CoCa), a new image-text foundation model family that subsumes existing vision pretraining paradigms with natural language supervision. Pretrained on image-text pairs from various data sources in a single stage, CoCa efficiently combines contrastive and captioning objectives in an encoder-decoder model. CoCa obtains a series of state-of-the-art performance with a single checkpoint on a wide spectrum of vision and vision-language problems. Our work bridges the gap among various pretraining approaches and we hope it motivates new directions for image-text foundation models.

**Broader Impact Statement**

This work presents an image-text pretraining approach on web-scale datasets that is capable of transferring to a wide range of downstream tasks in a zero-shot manner or with lightweight finetuning. While the pretrained models are capable of many vision and vision-language tasks, we note that our models use the same pretraining data as previous methods (Jia et al., 2021; Zhai et al., 2021a;b; Pham et al., 2021a) and additional analysis of the data and the resulting model is necessary before the use of the models in practice. We show CoCa models are more robust on corrupted images, but it could still be vulnerable to other image corruptions that are not yet captured by current evaluation sets or in real-world scenarios. For both the data and model, further community exploration is required to understand the broader impacts including but not limited to fairness, social bias and potential misuse.

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

## A    Visual Recognition Finetuning Details

| | ImageNet | | Kinetics-400/600/700 | | Moments-in-Time | |
|---|---|---|---|---|---|---|
| **Hyper-parameter** | Frozen-feature | Finetuning | Frozen-feature | Finetuning | Frozen-feature | Finetuning |
| Optimizer | Adafacter with Decoupled Weight Decay | | | | | |
| Gradient clip | 1.0 | | | | | |
| EMA decay rate | 0.9999 | | | | | |
| LR decay schedule | Cosine Schedule Decaying to Zero | | | | | |
| Loss | Softmax | | | | | |
| MixUp | None | | | | | |
| CutMix | None | | | | | |
| AutoAugment | None | | | | | |
| RepeatedAugment | None | | | | | |
| RandAugment | 2, 20 | 2, 20 | None | None | None | None |
| Label smoothing | 0.2 | 0.5 | 0.1 | 0.1 | 0.0 | 0.0 |
| Train steps | 200k | 200k | 120k | 120k | 120k | 120k |
| Train batch size | 512 | 512 | 128 | 128 | 128 | 128 |
| Pooler LR | 5e-4 | 5e-4 | 5e-4 | 5e-4 | 5e-4 | 5e-4 |
| Encoder LR | 0.0 | 5e-4 | 0.0 | 5e-4 | 0.0 | 5e-4 |
| Warm-up steps | 0 | 0 | 1000 | 1000 | 1000 | 1000 |
| Weight decay rate | 0.01 | 0.01 | 0.0 | 0.0 | 0.0 | 0.0 |

Table 8: Hyper-parameters used in the visual recognition experiments.

In addition to zero-shot transfer, we evaluate frozen-feature and finetuning performance of CoCa on visual recognition tasks. For frozen-feature evaluation, we add an attentional pooling layer (pooler) on top of the output sequence of visual features and an additional softmax cross entropy loss layer to learn classification of images and videos. For finetuning, we adapt the same architecture as frozen-feature evaluation (thus also with poolers) and finetune both encoder and pooler. All learning hyperparameters are listed in Table 8.

## B    Multimodal Understanding Finetuning Details

| **Hyper-parameter** | **VQA** | **SNLI-VE** | **NLVR2** | **MSCOCO** | **NoCaps** |
|---|---|---|---|---|---|
| Optimizer | Adafacter with Decoupled Weight Decay | | | | |
| Gradient clip | 1.0 | | | | |
| LR decay schedule | Cosine Schedule Decaying to Zero | | | | |
| RandAugment | 1, 10 | 1, 10 | None | None | None |
| Train steps | 100k | 50k | 50k | 50k | 10k |
| Train batch size | 64 | 128 | 64 | 128 | 128 |
| Pooler LR | 5e-4 | 1e-3 | 5e-4 | NA | NA |
| Encoder LR | 2e-5 | 5e-5 | 2e-5 | 1e-5 | 1e-5 |
| Warm-up steps | 1000 | 1000 | 1000 | 1000 | 1000 |
| Weight decay rate | 0.1 | 0.1 | 0.1 | 0.1 | 0.1 |

Table 9: Hyper-parameters used in the multimodal experiments.

CoCa is an encoder-decoder model and the final decoder outputs can be used for multimodal understanding/-generation. Thus, we evaluate on popular vision-language benchmarks. We mainly follow the same setup introduced in Wang et al. (2021b). All hyper-parameters are listed in Table 9.

For multimodal classification, we feed the image into the encoder and the corresponding text to the decoder. We then apply another attentional pooler with a single query to extract embedding from the decoder output, and train a linear classifier on top of the pooled embedding. For VQA v2 (Goyal et al., 2017), we follow prior work and formulate the task as a classification problem over 3,129 most frequent answers in the training set.

| | MSR-VTT Full | | | | | |
| | Text → Video | | | Video → Text | | |
| Method | R@1 | R@5 | R@10 | R@1 | R@5 | R@10 |
|---|---|---|---|---|---|---|
| CLIP (Portillo-Quintero et al., 2021) | 21.4 | 41.1 | 50.4 | 40.3 | 69.7 | 79.2 |
| Socratic Models (Zeng et al., 2022) | - | - | - | 44.7 | 71.2 | 80.0 |
| CLIP (Portillo-Quintero et al., 2021) (subset) | 23.3 | 44.2 | 53.6 | 43.3 | 73.3 | **81.8** |
| Socratic Models (Zeng et al., 2022) (subset) | - | - | - | 46.9 | **73.5** | 81.3 |
| CoCa (subset) | **30.0** | **52.4** | **61.6** | **49.9** | 73.4 | 81.4 |

Table 10: Zero-shot Video-Text Retrieval on MSR-VTT Full test set.

We additionally enable cotraining with the generative loss on the concatenated pairs of textual questions and answers to improve model robustness. Similarly for SNLI-VE, the image and the textual hypothesis are fed to encoder and decoder separately, and the classifier is trained to predict the relation between them as entailment, neutral or contradiction. For NLVR2, we create two input pairs of each image and the text description, and concatenate them as input to the classifier. We do not use image augmentation for NLVR2.

For image captioning, we apply simple cross-entropy loss (same as the captioning loss used in pretraining) and finetune the model on the training split of MSCOCO to predict for MSCOCO test split and NoCaps online evaluation. We use beam search with beam size of 4 for all our experiments.

## C Zero-Shot Video Retrieval

We evaluate video-text retrieval using CoCa on MSR-VTT (Xu et al., 2016) using the full split. Table 10 shows that CoCa produces the highest retrieval metrics for both text-to-video and video-to-text retrieval. It is important to note that MSR-VTT videos are sourced from YouTube, and we require the original videos to compute our embeddings. Many of the videos have been made explicitly unavailable (Smaira et al., 2020), hence we compute retrieval over the subset of data that is publicly available at the time of evaluation. Using code[1] provided by the authors of Socratic Models (Zeng et al., 2022), we re-computed metrics on the available subset for those methods, indicated by "(subset)" for fairest comparison.

---

[1] https://socraticmodels.github.io

