# OpenReview forum: "CoCa: Contrastive Captioners are Image-Text Foundation Models"
_TMLR — Accepted by TMLR_

### Review · Reviewer_VEHW · 2022-07-18

**Summary Of Contributions:**

In general, the paper brings a new way to combine the two existing methods: generative pre-training and contrastive pre-training. I am impressed by the ways that the method showing good qualitative results on downstream tasks and better training efficiency. Here is a summary of the contributions:

1. Proposing the contrastive captioner method, which combines generative pre-training and contrastive pre-training. The model design splits the language model into two parts: a unimodal text decoder and a multimodal text decoder. Although splitting decoder has been seen in previous works (by specifying the layers with cross-attentions), the exact way of using it to utilize a single unimodal feature (for contrastive learning) and next-word logits (for conditional language modeling) is novel to me.

2. As a unified model, the paper shows strong performance on different types of tasks, including visual recognition (classification), multimodal retrieval, and multimodal understanding. The adaptation to these downstream tasks are natural and achieve sota results on them.

3. The model is proved at large scale. Usually, the combination of two kinds of training paradigm is hard to survive in large scale since one paradigm might overshadow the other. Thus, showing the success at large scale can entail that these two tasks are complementary to each other.

4. A detailed ablation studies are provided regarding some key designs of the model, such as whether the captioning less can used to replace the classification loss.

**Broader Impact Concerns:**

I do not see any.

**Requested Changes:**

I think that the paper looks good to me and I have several questions regarding the details. All these questions are considered as minor to me and they would not affect my decision to this paper.

1. The description regarding the "learnable CLS tokens" (about Table 7 (e)) is not clear enough to me. I am not sure what the paper means by the four variations in Table 7 (e). The connection from the "unimodal decoder layers" to "text embedding design" is also not obvious given the text in "Unimodal and multimodal decoders" paragraph. I would suggest a rewriting here.

2. In Sec 3.3, "we also encode the captions of each video as target embeddings" is not clear to me. Is it the standard retrieval pipeline that encodes the text to embeddings and do nearest neighbor search?

3. There are several questions regarding the attention pooler. (a) What is the number of queries when applying to the image classification task? Is it a feature_map --> attention pooler with n_cls queries --> softmax NLL loss design or a feature_map --> attention pooler with 1 query --> FC to n_cls --> softmax NLL loss design? (b) In Figure 4.a, there is a comparison between CoCa and other works on ImageNet top-1 accuracy. Since CoCa uses attention pooler in its classification head, I would like to know the classification head for other works such as SwinV2, ViT, CoAttNet. (c) At the end of sec 4.3 and Table 7.f, it shows that using attention pooling of 256 queries shows similar results as 0 quires (thus directly using the original visual features). Sec 3.2 shows that the visual feature numbers are 16x16=256, thus I would like to see more explanation about the need of an additional attentional pooler to 256.

4. Does the image captioning evaluation use beam search or just greedy decoding? What the number of beams if so.

5. Missing citations for the decoupled weight decay paper, "Decoupled Weight Decay Regularization" in Sec 4.1, "optimization".




**Strengths And Weaknesses:**

## Strengths
1. The paper proposes the contrastive captioning method that naturally merges two pre-training frameworks for multimodal research. The split-decoder design can bring the contrastive learning and generative pre-training together without corrupting the input data. Another benefit of this specific design is that it can be easily adapted to different kinds of tasks, e.g., it does not require to repurpose the models for captioning tasks (OSCAR needs to shift the tokens for generative tasks, CLIP needs another training round).

2. The paper shows a very strong results on diverse types of tasks, such as image recognition, video recognition, multimodal retreival, multimodal text generation, and multimodal understanding. The paper also present results under different experimental setups, e.g., zero-shot, frozen-feature adaptation, and full fine-tuning. This quantitive evaluation on downstream tasks is comprehensive. Btw, I like the figure style of Fig.6, which is clear and tidy.

3. The paper shows the benefit of multitasking, which is not obvious according to previous works. For examples, in VilBERT 12-in-1, the separate trained models still show stronger performance than the multitask model. Thus CoCa provides a good examples that a careful design of task combination can bring significant results over original models.

4. Although the ablation studies of the paper is not exhaustive, they include most of the bullet that I care about. Some results are also inspiring and worth reading. For example, the paper shows that the captioning loss with comma-separated labels can be an alternative to the standard softmax classification loss. To the best of my knowledge, I did not see such results in previous literature.

5. The paper provides a comprehensive list of hyper-parameters used in the paper, which would help reproduction of the results in the paper.


## Weaknesses

The work is mostly complete to me and here are some improvements that I can think of.

0. No code/model releasing plan is specified in the paper, which I considered as a loss to the community. It would be great if a smaller model trained on public dataset can be released.

1. The pre-training of the model takes both generative tasks and contrastive tasks in a single run. However, in the downstream evaluation, only one of these tasks are utilized for most of the cases. It might be better to consider both tasks in downstream as well. For example, the image classification tasks can be formulated as a text-matching problem (as demonstrated in this paper) but can also be reformulated as a class-name generation problem, thus the later formulation can effectively leverage the. The same to the VQA task, where only discriminative approach is considered. Utilizing the generative modules may also extend the work to (semi-) open-domain VQAs, such as text-vqa, OK-VQA.

2. The visual tasks with spatial localization is not evaluated in this paper. I would think that the model would have the capacity to handle the tasks like object detection/segmentation since the cross-attention layer is included. A plain ViT adaption (like [1]) to detection tasks would be a great improvement to this work.

3. As a multimodal model, the model is proposed in an asymmetrical way where the specific designs are on the language sides. I would think that these designs can also be brought to the vision side by utilizing a similar framework.

[1] Li, Yanghao, Hanzi Mao, Ross Girshick, and Kaiming He. "Exploring plain vision transformer backbones for object detection." arXiv preprint arXiv:2203.16527 (2022).

---

> ### Author Response · Authors · 2022-07-26
> **Response to Reviewer VEHW**
>
> Thank you for your comprehensive review and valuable feedback! We address your comments one by one as following:
>
> [Combining Contrastive with Generative for Downstream]
> We agree this is a valuable direction to explore, as shown in [1, 2]. That said, generative approach is generally less efficient for zero-shot evaluation as it requires O(N^2) computation versus O(N) for retrieval-based approach. And thus we mainly consider retrieval-based methods for efficiency.
>
> [Spatial Localization]
> Thank you for your advice. We also believe the full potential of applying CoCa as the pretrained backbone for localization tasks and we leave it as a future work.
>
> [Asymmetrical Design]
> One of our key objectives is to train a strong visual foundation model and thus we chose to keep the design simple for fair comparison with our existing baselines. We also agree that similar designs can be applied on the visual side and we will explore in a future work.
>
> [In Sec 3.3…]
> Yes, it means encoding each caption into a unimodal text embedding independently. We have added this in the paper.
>
> [Attentional Pooler]
> Thank you for your detailed questions. (a) We use the 'feature_map --> attention pooler with n_cls queries --> softmax NLL loss' with 256 queries (same as pretraining). (b) In Figure 4.a we counted all parameters including classifier heads. For other networks, we obtain numbers from their corresponding publications and thus standard linear classification heads are used. (c) Although they both have 256 tokens during pretraining, using an attentional pooler will also restrict the number of tokens during finetuning on larger image resolutions, and thus is computationally much more efficient.
>
> [Image Captioning Evaluation]
> We use beam search with a beam size of 4 (added in the appendix).
>
> [Writing Suggestions]
> Thank you for your kind advice! We have addressed them in the manuscript accordingly.
>
> [1] Unifying Vision-and-Language Tasks via Text Generation. Cho et al., ICML 2021.
> [2] SimVLM: Simple Visual Language Model Pretraining with Weak Supervision. Wang et al., ICLR 2022.

---

> > ### Comment · Reviewer_VEHW · 2022-07-26
> > **Response to Authors**
> >
> > Thanks for the author's response. The response resolves my questions and concerns, and I am satisfied with the paper update.

---

### Review · Reviewer_u8gt · 2022-07-20

**Summary Of Contributions:**

This paper proposed a new method for image-text pretraining. Particularly, the proposed CoCa exploits an encoder-decoder architecture and combines contrastive and captioning objectives. On several downstream tasks, CoCa achieved state-of-the-art performances with zero-shot transfer or minimal task-specific adaptation.

**Requested Changes:**

1. The ablation studies show that the contrastive loss helps little for the VQA task and has not shown it can help ZS. The paper should add more experiments to demonstrate the effectiveness of the contrastive loss.

2. The paper missed some related and important literatures. They have to be cited and compared.
a. An empirical study of gpt-3 for few-shot knowledge-based vqa. A
b. A Good Prompt Is Worth Millions of Parameters? Low-resource Prompt-based Learning for Vision-Language Models
c. Multimodal Few-Shot Learning with Frozen Language Models
d. Unifying Vision-and-Language Tasks via Text Generation

**Strengths And Weaknesses:**

1. The new architecture omits cross-attention in the first half of decoder layers to encode unimodal text representations and cascades the remaining decoder layers that cross-attend to the image encoder for multimodal image-text representations, which is efficient for training.

2. The training objectives jointly combine contrastive loss and captioning loss, taking both benefits from contrastive approaches and generative approaches.

3. The empirical results are good.

---

> ### Author Response · Authors · 2022-07-26
> **Response to Reviewer u8gt**
>
> Thank you for your comprehensive review and valuable feedback! We address your comments one by one as following:
>
> [Benefit for Contrastive Loss]
> In fact, the zero-shot crossmodal retrieval (including zero-shot image classification) capability is enabled by contrastive loss, and thus no score is reported for the captioning-only model. We have also noted it in our discussion.
>
> [Missing References]
> Thank you for your pointers. We have added these in the updated manuscript.

---

### Review · Reviewer_gXWo · 2022-07-21

**Summary Of Contributions:**

This paper introduces a simple combination of contrastive loss and captioning loss to pre-train an image-text foundation model, with a ender-decoder architecture, dubbed CoCa. To make contrastive loss to work along with captioning loss,  CoCa keeps the first half of the decoder for unimodal language encoding and the later half for multimodal fusion. CoCa can be viewed as an unification of previous work CLIP and SimVLM in both model architecture and pre-training losses. This simple unification enables CoCa to achieve superior performance on a wide range of downstream tasks with zero-shot transfer or minimal task-specific adaptation, including image/video classification, multimodal retrieval/understanding and image captioning.

**Broader Impact Concerns:**

Discussions about some potential societal impact are included in the paragraph following the conclusion section, which I believe is sufficient.

**Requested Changes:**

- For the comparison of training costs in table 7b, does the captioning only model (referred as naive encoder-decoder architecture in the discussion paragraph) contain a unimodal text decoder? If not, how may layers are used in multimodal decoder for this captioning model?


**Strengths And Weaknesses:**

Strengths:

Overall, this was a good read for me. The paper is clearly written and easy to follow. The presentation of method and results are also in a good flow. The evaluation of the proposed method is also quite sufficient. The model has been empirically shown to achieve strong performance over many downstream tasks, with abundant ablation studies to verify the design choices of the final model.

Weakness:

No major weakness, please refer to section below for some minor questions.

---

> ### Author Response · Authors · 2022-07-26
> **Response to Reviewer gXWo**
>
> Thank you for your comprehensive review and valuable feedback! We address your comments one by one as following:
>
> [Captioning Model Architecture]
> The captioning-only model uses the same architecture as CoCa and thus contains 12 layers in the decoder in total (6 unimodal layers and 6 multimodal layers). This is because we found a 12 multimodal layers variant to perform similarly, and thus uses this version to ensure a fair comparison for computational efficiency. We have updated the manuscript.

---

### Decision · Action_Editors · 2022-08-17

**Recommendation:** Accept with minor revision

**Comment:**


This paper introduces Contrastive Captioners (CoCa), a new family of multi-modal foundation models. CoCa enables large scale multi-modal pre-training and combines generative pre-training and contrastive pre-training. In terms of architecture, Coca subsumes single-encoder, dual-encoder and encoder-decoder paradigms, and uses a decoupled decoder without any cross-attention in the unimodal text decoder.

The submission received positive reviews, with two accept recommendations and one leaning accept recommendation. The reviewers pointed out the relevance of the combination of the generative and contrastive pre-trainings, as well as the decoupled decoder architecture. The claims are also strongly experimentally validated, since the approach reaches very good performances on several downstream tasks, *e.g.* visual recognition (classification), cross-modal alignment, image captioning and multimodal understanding. Numerous ablation studies also validate the proposed design choices.
The authors successfully answered to the main reviewers concerns during the discussion period.

The AE carefully reads the submission and agrees that the decoupled decoder architecture is novel and that the experimental validation is strong.
The AE thus recommends paper acceptance under the two minor changes/answers:
- The references provided by Ru8gt (Yang et al., 2022b; Jin et al., 2021; Tsimpoukelli et al., 2021) have been included in the related work section ; it would be nice to position Coca more precisely with respect to these recent baselines.
- RVEHW raised the question of the diffusion of open source code and of model release. The AE indeed considers that it would be an important added valued to the community.

---

> ### Author Response · Authors · 2022-08-23
> **Camera Ready Revision**
>
> We thank all reviewers for their time and efforts, and AE for your careful reviews!
>
> We have uploaded our revised camera-ready version with comments addressed. We would like to thank AE and reviewer VEHW for the clarification of the difference between open-sourcing code and releasing models. We totally agree on the value of open-sourcing the implementation (in fact our Figure 2 Algorithm 1 has a pseudo code of CoCa itself). We have also started the code release process and will update once they are ready. Before that, we'd also be happy to provide detailed information for any question related to implementation of CoCa.